# Defining the Cynomolgus Macaque (*Macaca fascicularis*) Animal Model for Aerosolized Venezuelan Equine Encephalitis: Importance of Challenge Dose and Viral Subtype

**DOI:** 10.3390/v15122351

**Published:** 2023-11-29

**Authors:** Crystal W. Burke, Christina L. Gardner, Aimee I. Goodson, Ashley E. Piper, Rebecca A. Erwin-Cohen, Charles E. White, Pamela J. Glass

**Affiliations:** 1Virology Division, U.S. Army Medical Research Institute of Infectious Diseases, Frederick, MD 21702, USAaimee.i.goodson.civ@health.mil (A.I.G.);; 2Statistics Division, U.S. Army Medical Research Institute of Infectious Diseases, Frederick, MD 21702, USA; 3Risk Management Office, U.S. Army Medical Research Institute of Infectious Diseases, Frederick, MD 21702, USA

**Keywords:** Venezuelan equine encephalitis virus, cynomolgus macaque, animal model, aerosol exposure, medical countermeasures, nonhuman primates, study design

## Abstract

Venezuelan equine encephalitis virus (VEEV) outbreaks occur sporadically. Additionally, VEEV has a history of development as a biothreat agent. Yet, no FDA-approved vaccine or therapeutic exists for VEEV disease. The sporadic outbreaks present a challenge for testing medical countermeasures (MCMs) in humans; therefore, well-defined animal models are needed for FDA Animal Rule licensure. The cynomolgus macaque (CM) model has been studied extensively at high challenge doses of the VEEV Trinidad donkey strain (>1.0 × 10^8^ plaque-forming units [PFU]), doses that are too high to be a representative human dose. Based on viremia of two subtypes of VEEV, IC, and IAB, we found the CM infectious dose fifty (ID_50_) to be low, 12 PFU, and 6.7 PFU, respectively. Additionally, we characterized the pattern of three clinical parameters (viremia, temperature, and lymphopenia) across a range of doses to identify a challenge dose producing consistent signs of infection. Based on these studies, we propose a shift to using a lower challenge dose of 1.0 × 10^3^ PFU in the aerosol CM model of VEEV disease. At this dose, NHPs had the highest viremia, demonstrated a fever response, and had a measurable reduction in complete lymphocyte counts—biomarkers that can demonstrate MCM efficacy.

## 1. Introduction

Alphaviruses are enveloped viruses that contain a single-stranded, positive-sense genome that is capped and polyadenlyated. The mosquito-vectored encephalitic alphaviruses cause periodic epizootic and epidemic outbreaks in horses and humans. Venezuelan equine encephalitis virus (VEEV) is the prototype member of the encephalitic alphaviruses and has been extensively studied since its identification in the early 1930s [1,2]. VEEV was previously developed as a biological threat agent [3] because it can be easily produced, is stable when lyophilized, requires a low infectious dose (human infection with an aerosol exposure dose of only 10–100 PFU [4]), and has a 1:1 symptomatic to asymptomatic ratio [5,6]. Natural VEEV infection typically results in an acute febrile illness with severe symptoms subsiding within 3–5 d; however, 14–20% of symptomatic individuals develop neurological symptoms. Despite the development of an acute, often debilitating, disease in humans, the mortality rate of VEEV is low, less than 1% [7,8,9]. Symptoms include fever, chills, malaise, severe headaches, photophobia, and myalgia, especially in the lumbosacral area and legs [10,11,12,13,14]. Gastrointestinal symptoms such as vomiting and diarrhea have also been reported [12,14].

Epizootic IAB and IC subtypes of VEEV have been responsible for most of the widespread epidemics and human cases of VEEV disease when compared to the endemic ID and IE subtypes also associated with human infection [15,16,17,18]. The IAB subtype viruses were responsible for outbreaks between 1938 and 1973; since that time, the majority of widespread epidemics have been caused by an IC subtype [4]. Despite its absence from the currently circulating VEEV subtypes, the VEEV subtype IAB has a history of development as a biological threat agent and thus remains an agent of concern for the chemical and biological defense communities [3]. For these reasons, efforts to develop well-characterized disease models to evaluate medical countermeasures (MCM) for both IAB and IC VEEV subtypes are warranted [4].

While equine vaccines are available to protect against encephalitic alphaviruses, there are currently no licensed vaccines or effective antiviral therapies to protect humans. The VEEV TC-83 vaccine was previously administered to at-risk laboratorians under FDA investigational new drug (IND) status as part of the USAMRIID’s Special Immunizations Program; however, this program was recently halted. The low incidence of naturally occurring human disease may necessitate FDA licensure of a VEEV vaccine or therapeutic under the Animal Rule. To develop a MCM under the Animal Rule, an adequate and well-controlled animal model with a well-documented natural history that reasonably reflects disease processes present during human infection must be available. The cynomolgus macaque (CM) has been used for efficacy testing of VEEV vaccines and therapeutics [19,20,21,22,23,24,25] because this nonhuman primate (NHP) develops disease signs that mimic human disease [26]. Many studies of MCMs to protect CMs from VEEV disease have used high-dose virus challenges (≥1.0 × 10^6^ PFU) by aerosol exposure [20,23,24,27,28]. However, it is unknown what, if any, disease is present when macaques are exposed to lower, more human exposure-relevant doses of VEEV [4].

Here, we conducted a series of studies to characterize VEEV disease in CMs exposed by small aerosol particles to INH-9813, the IC strain of VEEV isolated from human serum during a 1995 outbreak in Colombia [29], or Trinidad donkey (TrD), the IAB strain of VEEV obtained from a donkey brain isolate during a 1943 outbreak in Trinidad [30]. The objectives of these studies were twofold: (1) identify the lowest infectious dose by aerosol delivery to observe VEEV disease in CMs and (2) compare the ability of IC and IAB strains to produce readily apparent signs of infection (viremia, fever, and lymphopenia) in CMs that could be used for MCM development. To meet these objectives, we initially used a staircase study design that utilized up- and down dosing to identify the range of infectivity of VEEV INH-9813 in macaques. Next, follow-on studies of sufficient statistical power using larger groups of these NHPs at low, medium, and high exposure doses were performed in an attempt to identify a single clinical parameter that could consistently provide an indicator of a MCM’s efficacy. Finally, we performed a bridging study with CMs exposed to the IAB subtype of VEEV, TrD strain, at various doses to determine if the low infectivity and viremia, fever, and lymphopenia patterns identified after infection with VEEV INH-9813 were consistent amongst epizootic VEEV subtypes.

## 2. Materials and Methods

### 2.1. Ethics Statement

Research was conducted under an institutional animal care and use committee-approved protocol in compliance with the Animal Welfare Act, Public Health Service Policy, and other federal statutes and regulations relating to animals and experiments involving animals. The facility where this research was conducted is accredited by the Association for Assessment and Accreditation of Laboratory Animal Care International and adheres to principles stated in the Guide for the Care and Use of Laboratory Animals, National Research Council, 2011.

### 2.2. Study Design: Staircase Studies

Eleven NHPs, cynomolgus macaques (*Macaca fascicularis*), were exposed to increasing or decreasing doses of VEEV INH-9813 and monitored for disease. In the staircase ID_50_ determination method, if an NHP exhibited the markers of infection with the initial dose, the NHP in the subsequent step received a lower dose. This process continued by either increasing or decreasing the inhaled dose of the virus until the range between an infectious or noninfectious dose was identified using one NHP for each dose.

### 2.3. Study Design: Modified Refinement Study, INH-9813

At the outset of the experiment, infection parameters were expanded to include viremia, fever, lymphopenia, and the production of neutralizing titers. Twenty-four of the NHPs were randomized into three groups of eight animals each. Groups 1, 2, and 3 were aerosol exposed to low (target dose 10 PFU), medium (target dose 1.0 × 10^3^ PFU), or high (target dose 1.0 × 10^6^ PFU) doses of VEEV INH-9813, respectively.

### 2.4. Study Design: Modified Refinement Study, TrD

Sixteen of the NHPs were randomized into four groups of four animals each. Groups 1, 2, 3, and 4 received a target inhaled dose of 1.0 × 10^1^, 1.0 × 10^3^, 1.0 × 10^5^, and 1.0 × 10^7^ PFU of TrD, respectively. The NHPs were evaluated for indications of infection, including viremia, fever, lymphopenia, and the production of neutralizing titers.

### 2.5. Virus Stock Preparation: VEEV INH-9813 Strain

The VEEV INH-9813 strain passaged once in Vero cells (V-1) was generously provided by Dr. Robert Tesh at the University of Texas Medical Branch, Galveston, TX. A master virus stock (MVS) was prepared by infecting Vero 76 cells at a multiplicity of infection (MOI) of 1. Working virus stock (WVS) was prepared by infecting Vero 76 cells at a MOI of 1. Supernatants were harvested at 24 h post-infection, clarified by low-speed centrifugation, aliquoted, and stored at −80 °C. A sucrose-purified stock was prepared by infecting Vero 76 cells with MVS at a MOI equal to 1. Supernatants were harvested at 24 h post-infection, clarified by low-speed centrifugation, and purified through a 60/20 sucrose gradient followed by a 20% sucrose cushion. The sucrose stock was used for NHPs #1 and #2 in the staircase study, while the WVS was used for all other NHPs challenged with INH-9813. Mycoplasma, sterility, and endotoxin testing were performed on each stock. The WVS was deep-sequenced to confirm no contamination by other pathogens. The sucrose-purified stock was sequenced and published in GenBank [31].

### 2.6. Virus Stock Preparation: VEEV Trinidad Donkey Strain

A historical stock of the VEEV TrD strain (passages: 1 guinea pig, 13 eggs, and 1 duck embryonic cell) was used as the starting material. A MVS was prepared by infecting Vero 76 cells at a MOI of 1.5. WVS was prepared by infecting certified Vero 76 cells (ATCC) at a MOI of 2. Supernatants were harvested at 24 h post-infection, clarified by low-speed centrifugation, filtered through Amicon Ultra Centrifugal units (MilliporeSigma, Darmstadt, Germany) to concentrate, then aliquoted, and stored at −80 °C. This stock was characterized through mycoplasma, sterility, and endotoxin testing and was deep-sequenced to confirm identity.

### 2.7. Animals

CMs of Indochinese origin that were >4 years of age, ≥3 kg, alphavirus naïve, and free of specific pathogens (e.g., Salmonella, Shigella, Campylobacter, Klebsiella hypermucoviscosity phenotype (HMV) strain, tuberculosis, SIV, STLV1/2, SRV1/2/3, and Herpes B virus) as determined by vendor testing were used for these studies. These NHPs were randomized by sex and weight into groups by a statistician for each phase of the study, with the INH-9813 staircase study as an exception due to the nature of the study design. Hematology and clinical chemistry values for each animal were evaluated and deemed acceptable by a USAMRIID veterinarian for study acceptance. At least 14 d prior to the start of the study, the NHPs were implanted with telemetry implants (DSI M00, Harvard Bioscience, Inc. (Holliston, MA, USA), or ITS-T2J) for the collection of temperature data. Upon recovery, approximately 10 d prior to aerosol exposure, NHPs were transferred into the animal biosafety level-3 (ABSL-3) containment facility for acclimation and collection of baseline telemetry data. Blood was collected 3 days prior to the challenge to establish baseline hematology and clinical chemistry levels for each NHP. The aerosol exposure dose for each animal was calculated from the minute volume determined with a plexiglass whole-body plethysmograph box using either Buxco XA v2.9 or Buxco FinePoint v2.3.1.6 software. The total volume of aerosol breathed was determined by the exposure time required to deliver the estimated inhaled dose. Animals were exposed to a head-only automated bioaerosol exposure system (ABES-II). The exposure was generated using a collision nebulizer to produce a highly respirable aerosol (flow rate: 7.5 ± 0.1 L/min). The system generated a target aerosol of 1 to 3 μm mass median aerodynamic diameter as determined by a TSI Aerodynamic Particle Sizer (APS) spectrometer. During the aerosol exposure, all-glass impinger (AGI) samples were collected for titration of virus exposure and calculation of the actual inhaled dose for each NHP as previously described [25]. AGI samples were quantified by plaque assay. Blood was collected for hematology analysis and for the analysis of viremia by plaque assay and RT-PCR. At the study endpoint of 28 d after exposure, all NHPs were euthanized.

### 2.8. Clinical Observations

Baseline behavioral data were collected for each animal for comparison purposes. Awake observations were performed twice daily by study personnel for signs of clinical disease. Physical examinations were completed anytime the animals were anesthetized to assess dehydration, lymph node size, and body condition. Specific observations for the VEEV INH-9813 studies included: (1) animal responsiveness with and without stimuli; (2) neurological status, including changes in eye pattern movements, light sensitivity, vocalization, or perceived tremors; (3) behavior, including changes in level of activity or social interaction; and (4) changes in body temperature. For the VEEV TrD study, institute-directed changes in the awake clinical observation criteria resulted in reduced granularity of alphavirus-specific disease signs. Therefore, the only specific observations assessed were neurological abnormalities, including tremors and responsiveness to stimuli.

### 2.9. Hematology Analysis

Quantitative blood cell counts were conducted using a Cell Dyn 3700 (Abbott, Abbot Park, IL, USA) or VETSCAN HM5 (Abaxis, Union City, CA, USA) clinical hematology analyzer in accordance with the institute’s SOPs. Assays were completed within 2 h of blood collection. Complete blood count (CBC) parameters were measured using whole blood from EDTA tubes. Lymphopenia was defined as a ≥30% decrease in total lymphocyte counts when compared to the average of baseline values for each individual NHP.

### 2.10. Telemetry

Temperature data were collected continuously, starting at least 5 d prior to aerosol exposure, at a rate of one sample per second. These data points were automatically averaged and statistically filtered by the data collection software to remove noise and signal artifacts, and a single data point was generated every 30 s. Data were transferred to a validated Excel spreadsheet for analysis. All values corresponding to the same time of day during the baseline period were averaged with a standard deviation (SD) calculated to produce a normal baseline reference temperature table for each hour of the day during a 24 h time period. When considered appropriate (e.g., while the animals were sedated), temperature data points inconsistent with the other readings observed during that timeframe on the same day and across days were removed.

### 2.11. Plaque Assay

A plaque assay was used to assess infectious viruses in blood, nasal and throat swabs, and tissues processed following necropsy. Briefly, samples were serially log diluted starting at a dilution of 1:10 in Hank’s Balanced Salt Solution (HBSS), 2% heat-inactivated Fetal Bovine Serum (HI-FBS), 2% penicillin/streptomycin (Pen/Strep, 10,000 IU/mL/10,000 µg/mL), and 1% HEPES. Vero 76 cells seeded on 6-well plates were grown to ~90–100% confluence. Cells were infected with 0.1 mL of each serial dilution per well in duplicate. Plates were incubated at 37 ± 2 °C for 1 h ± 15 min with gentle rocking every 15 min. After 1 h, cells were overlaid with 1.2% agarose mixed in a ratio of 1:1 with Basal Medium Eagle (BME), containing 10% HI-FBS and 2% Pen/Strep, and incubated for 24 ± 4 h at 37 ± 2 °C and 5 ± 1% CO_2_. A second overlay containing 1.2% agarose mixed 1:1 with BME with 10% heat-inactivated FBS, 2% Pen/Strep, and 5% of total volume neutral red vital stain was added to the wells. They were further incubated for 18–24 h for the visualization of plaques. The AUC for viremia was determined using GraphPad Prism. Prism utilizes the trapezoid rule when calculating.

### 2.12. Plaque Reduction Neutralization Test (PRNT)

The levels of neutralizing antibodies present were measured using the plaque reduction neutralization test. Blood was collected into serum separator tubes, processed, and frozen at −60 to 80 °C until use. Prior to assessment, serum samples were heat inactivated for 30 ± 1 min at 56 ± 2 °C. Samples were serially diluted to a ratio of 1:2, starting at a 1:10 dilution, in the minimum essential media (MEM) complete medium (MEM with Phenol Red containing 1:10 HI FBS, 1:50 Pen/Strep, and 1:100 HEPES). Virus stocks were diluted to a concentration of 2.0 × 10^3^ PFU/mL and then added in a ratio of 1:1 to the diluted serum samples and negative (naïve NHP sera) or positive (anti-alphavirus antibody or anti-sera) controls. Samples were incubated overnight at 2–8 °C. ATCC Vero 76 cells seeded on 6-well plates were grown to ~90–100% confluence. Cells were infected with 0.1 mL of each serum + virus sample dilution per well in replicate wells. Plates were incubated at 37 ± 2 °C and 5 ± 1% CO_2_ for 1 h ± 15 min with gentle rocking every 15 ± 2 min. After incubation, cells were overlaid with 1.2% agarose mixed 1:1 with BME complete (BME containing 1:10 HI FBS and 1:50 Pen/Strep) and incubated for 24 ± 4 h at 37 ± 2 °C and 5% ± 1% CO_2_. A second overlay containing 1.2% agarose mixed 1:1 with BME complete with neutral red vital stain (1:20) was added to the wells. Cells were incubated at 37 ± 2 °C under 5 ± 1% CO_2_ overnight for visualization of plaques. The plaques were counted, and PRNT80 values were determined.

### 2.13. Semi-Quantitative Real-Time Reverse-Transcriptase Polymerase Chain Reaction (RT-PCR) Assay

RT-PCR was utilized to detect the presence or absence of viral RNA per mL of blood or per mg of tissue. These assays were developed as a diagnostic assay for alphavirus infection. As such, the RT-PCR assay was not intended to be quantitative in nature. Experimental conditions were modified for the semi-quantitative analysis of RNA copies in samples by comparing blood and tissue samples to a linear synthetic RNA standard. Viral RNA was extracted from blood and tissues using commercially available kits (QIAmp Viral RNA kits, Qiagen, Hilden, Germany) according to the manufacturer’s specifications. Positive and negative extraction controls (PEC and NEC, respectively) were created by supplementing uninfected control NHP blood with a known amount of virus (1.0 × 10^4^ PFU for the PEC) or RNase-free water (for the NEC). The primers used for IC studies were forward, CTGTTTAAGCTTGGCAAACC; reverse, AATTCCCACTCGATTCCAGC; probe, 6FAMTGACAGGAGAAGGGCATTACACGAAGAGMGBNFQ. Primers for the IAB studies were forward, CTGTTTAAGCTTGGCAAACC; reverse, ATACCCACTCGGTTCCAGCG; and probe, 6FAMTGACAGGAGAAGGGCATTGCATGAAGAGMGBNFQ. The assay limit of detection ranged from 1.0 × 10^7^ viral genome copies/µL (upper limit of detection [ULOD]) to 1.0 × 10^1^ viral genome copies/µL (lower limit of detection [LLOD]).

### 2.14. Statistical Analyses

Missing data were not replaced, nor were individual missing values within any subject’s record imputed. Missing data were handled as missing at random, and no corrections for missing data were included in the analysis. Where measurements of any type were reported below the limit of detection, measurement results were set equal to the LLOD. An exact Wilcoxon test [5] was used to compare the median difference in measurement results between dose groups. The level at which statistical significance is achieved (alpha) was set to less than or equal to 0.05. Statistically significant two-sided tests are equivalent to including or not including zero in a 95% confidence interval (two-tailed).

## 3. Results

### 3.1. VEEV INH-9813 Is Extremely Infectious through Aerosol Exposure

Much of the data in the literature about VEEV disease is derived from vaccine efficacy studies [20,21,23,24,27,32] in animal models. Historically, these studies used high challenge doses (~1.0 × 10^8^ PFU) of the VEEV TrD strain because it was thought that protection from the highest possible aerosol dose of virus would be sufficient to demonstrate efficacy. To this end, little is known about the VEEV disease manifestations when animals are exposed to aerosols at lower doses of the virus. Here, we aimed to establish the minimum infectious dose of VEEV INH-9813 in NHPs. Eleven CMs (NHP 1 to NHP 11) were exposed to small aerosol particles in a staircase study with doses ranging from 30 to 5.94 × 10^8^ PFU (Table 1). Nasal and throat swabs were collected directly after aerosol exposure as a confirmation of viral exposure. The virus was detected using a plaque assay from these swabs in only those animals exposed to doses >1.0 × 10^4^ PFU, likely due to the overall lower concentration of virus delivered.

Blood samples were collected 3 d before and 1–10, 12, 14, 16, 18, 21, 24, and 28 d after exposure to evaluate viremia and absolute blood counts. Temperature changes were continuously monitored through implanted telemetry devices. All animals, regardless of dose, displayed classical markers of VEEV infection, including viremia, fever, and lymphopenia (Figure 1 and Table 1). Viremia was present for at least two and at most four consecutive days in the animals (Figure 1A). However, the number of days of viremia was not correlated with the dose of virus presented to the animal, as both the animal receiving the highest and lowest challenge dose had 3 d of viremia (NHP 2 and NHP 11, respectively). In general, animals receiving higher challenge exposures (>5.0 × 10^4^ PFU) had viremia detected 24 h after exposure, whereas those receiving lower doses (~1.0 × 10^4^ PFU or less) did not have detectable viremia until 48–72 h after exposure. Unexpectedly, the magnitude of infection, measured using the area under the viremia curve (AUC), was not correlated with the aerosol exposure dose (R^2^ < 0.01; Figure 2A).

The temperature response in rhesus [33] and CMs [34] after exposure to VEEV TrD has been described as a biphasic febrile illness [35]. For this reason, fever, defined as a temperature greater than 3 SD above the baseline value during a matched 30 min interval in a 24 h time frame, was measured in all animals (Appendix A). In general, animals exposed to higher challenge doses had greater fever-hours (Figure 1B); however, this was not always the case, as NHP 7 (6.75 × 10^4^ PFU) had greater fever-hours than the animals receiving the highest challenge doses, suggesting only a weak correlation between challenge dose and the strength of the host fever response (R^2^ = 0.4587; Figure 2B). NHP 11, which was exposed to just 30 PFU, had a very mild fever with only 32.8 fever-hours measured across the 25 d with available data; the telemetry device in this animal did not function for approximately 24 h around day 11 and stopped functioning on day 26 after exposure, long after any fever response had subsided.

Absolute lymphocyte counts were measured, and the percent change from baseline levels was calculated for each day of samples collected (Figure 1C). Lymphopenia, a greater than 30% reduction in the absolute lymphocyte counts, is denoted by the red dotted line. All NHPs had at least 1 d of lymphopenia, with all but NHP 7 having 3 or more days of lymphopenia. Notably, animals exposed to ≥1.0 × 10^6^ PFU had lymphopenia as early as 24 h after exposure, while lymphopenia was not present until 48–72 h after exposure in animals exposed to lower doses of VEEV INH-9813. Taken together, these data demonstrate that aerosol exposure of NHPs to even low doses of VEEV INH-9813 results in the development of viremia and clinical manifestations (fever and lymphopenia), as has been seen following accidental laboratory exposures in humans [26].

### 3.2. Refinement of the VEEV Disease Markers after Low, Medium, or High Exposure Doses

To refine the model further, three groups of animals (n = 8 each) were aerosol exposed to either low (target dose 10 PFU), medium (target dose 1.0 × 10^3^ PFU), or high (target dose 1.0 × 10^6^ PFU) doses of VEEV INH-9813 in an effort to characterize disease at a given VEEV exposure dose with statistical power. An abundance of data is available about VEEV disease when animals are exposed to 1.0 × 10^8^ PFU as that dose has been repeated in previous vaccine studies [20,21,23,24,27,32]. For that reason, we chose the 1.0 × 10^6^ PFU dose as the highest dose in order to build a body of data at this lower dose that has been reported previously as a low dose exposure [36]. NHPs in Group 1 received calculated inhaled doses ranging from 14.7 to 27.8 PFU, with an average of 22.9 PFU. NHPs in Group 2 received calculated inhaled doses ranging from 358.7 to 684.8 PFU, with an average of 570.4 PFU. Finally, NHPs in Group 3 received calculated inhaled doses ranging from 9.06 × 10^5^ to 2.46 × 10^6^ PFU, with an average of 1.32 × 10^6^ PFU. Individually calculated inhaled doses are found in Table 2.

This study was performed in two iterations, with Group 1 and 2 animals (n = 16 total) exposed to aerosol on a single day and monitored for disease for 28 d. Animals in Group 3 (n = 8) were exposed at a later date and monitored for 28 d. All animals were implanted with telemetry devices to monitor fluctuations in temperature over the course of the study. Blood samples were collected 3 d prior to the challenge to establish baseline values and on days 1–7, 14, 21, and 28 after aerosol exposure to monitor viremia and complete blood counts. Clinical observations were made prior to exposure to establish baseline behaviors for each animal, and observations were made twice a day after the aerosol challenge.

Plaque assays were performed on samples collected on days 1–7 after exposure to determine the levels of viremia (Figure 3A–C) in each group. In the group exposed to the lowest aerosol dose, viremia was only detected in two animals (NHP 1 and NHP 6) by plaque assay (Figure 3A). These animals received a calculated inhaled dose of 20 PFU and 36 PFU, respectively. While NHP 1 only had detectable viremia on day 3, NHP 6 had detectable viremia for 3 consecutive days (days 3–5). Despite this, viral genomic RNA was detected by RT-PCR in the blood of all animals in the low-dose group except NHP 7 (Table 2). All animals in the middle-dose, 1.0 × 10^3^ PFU, group had detectable viremia by plaque assay and RT-PCR (Figure 3B and Table 2). Peak viremia titers in this group ranged from 1.0 × 10^3^ to 1.2 × 10^7^ PFU/mL (mean 2.6 × 10^6^ PFU/mL, 1.4 × 10^6^ PFU/mL SEM; Figure 4), demonstrating how individual host responses can influence the development of viremia as the calculated inhaled doses in this group had nominal variation (1.1 × 10^3^ to 1.9 × 10^3^ PFU). Beyond peak titers, the number of days that viremia was detected in NHPs in this group also varied from a single day (NHP 15) to up to 4 d (NHP 9, NHP 12, and NHP 16). Viremia in the high-dose group was detected in 7 of the 8 NHPs. Viremia curves were more consistent in this group, with viremia detected in the 7 animals starting 1 d after aerosol exposure and lasting until day 4 or 5 after exposure. The peak titers in this group were less variable than the other groups, ranging from 4.1 × 10^3^ to 7.1 × 10^4^ PFU/mL. Interestingly, despite exposure to a higher aerosol dose, the mean maximum viremia value for the 1.0 × 10^6^ PFU group was approximately 100-fold lower (3.1 × 10^4^ PFU/mL, 8.4 × 10^3^ PFU/mL SEM) than the 1.0 × 10^3^ PFU group (Figure 4). Viremia, as measured by plaque assay, was not detected in NHP C02; however, genomic RNA was detected on day 3 by RT-PCR, suggesting levels of viremia below the plaque assay limit of detection (50 PFU/mL).

Next, differences in body temperature alterations after exposure to low, medium, or high doses of VEEV INH-9813 were examined. Out of the 8 animals in the low-dose group, one animal, NHP 6, experienced elevated temperatures on day 2, which peaked on day 3 after exposure (Figure 3D and Appendix A), corresponding with the initial detection of viremia by plaque assay (Figure 3A). In the low-dose group, NHP 1 also had detectable temperature alterations (by measuring fever-hours) (Figure 3D and Table 2) and was the only other NHP in the group to have detectable viremia by plaque assay. All NHPs in the medium-dose group had measurable temperature alterations, although the intensity and duration of the fever were lower in NHP 14, occurring only on days 6 to 7, which corresponded with when viremia was observed in this animal (Figure 3E). A clear biphasic fever response was observed in some animals (NHPs 9, 10, 11, and 13) but was absent or difficult to distinguish in others (NHPs 12, 14, 15, and 16; Appendix A). Mirroring the trend observed with viremia, the median and range (473; 186.3–880.1) of total fever-hours measured for NHPs in the medium-dose group were greater than those of the NHPs in the high-dose group (380.3; 156.8–783.6). Fever in the high-dose group was observed more rapidly than the other dose groups, coinciding with the development of viremia just 24 h after exposure (Figure 3F). NHP C02 was an exception, as fever wasn’t observed until 3 d after exposure. This animal also did not develop a measurable viremia by plaque assay (Figure 3C), but low levels of genomic RNA could be detected by RT-PCR on days 2, 3, and 4 (Table 2). This suggests that late fever onset may be due to a low-level infection only detectable by RT-PCR. Like the middle-dose group, some NHPs (C01, C04, C08, and C09) displayed biphasic fever while other NHPs (C02, C05, C06, and C07) did not (Appendix A).

For the majority of animals in the low-dose group, complete lymphocyte counts remained stable after exposure. Reduced lymphocyte counts were noted in two animals (NHP C06 and NHP C03); however, lymphopenia was only observed in NHP C06, and it coincided with the first day viremia was observed (Figure 3G). Lymphopenia was present on at least 1 d in all animals in both the middle- and high-dose groups (Figure 3H and Figure 3I, respectively). In the middle-dose group, NHP 15, which had a single day of measurable viremia by plaque assay, also only had 1 d of lymphopenia. NHP C02 in the high-dose group had 3 d of lymphopenia but no measurable viremia by plaque assay. The average duration of lymphopenia was 3 d for the middle-dose group and 4 d for the high-dose group (Table 2).

### 3.3. Clinical Observations of VEEV INH-9813 Disease across Exposure Doses

An examination of blinded clinical observations made across all 35 NHPs in these studies revealed a pattern of clinical signs associated with aerosol exposure to VEEV INH-9813 (Table 3).

The most common clinical sign observed in 94% of the animals was tremors, predominately of the extremities, that were most notable during movement. Upon physical examination, lymphadenopathy was noted in 91% of the NHPs, including all (100%) of the females and 87% of the males studied. In the majority of the animals, loss of appetite (63%) and lethargy (57%) were also observed. Despite an apparent loss of appetite defined by a reduction in biscuit consumption, weight loss after exposure was not observed (Appendix A). In approximately one-third of the animals, alterations in grooming (either piloerection or lack of grooming), dehydration, and nystagmus were recorded. Other less common clinical signs observed included hyperactivity/reactivity, teeth baring, depressed countenance, aggression, vomiting, diarrhea, and photophobia.

### 3.4. Infectious Dose Fifty (ID_50_) of VEEV INH-9813 Based on Viremia, Fever, and Lymphopenia

Beyond characterizing the disease progression and signs of infection after VEEV INH-9813 infection, the secondary goal of the above studies was to identify the ID_50_ of this strain in the CM model. Bayesian probit analysis of the three hallmarks of VEEV disease found the ID_50_ to be 12 PFU when viremia is used as a marker of infection, 25 PFU when fever is used as a marker, and 83 PFU when using lymphopenia as a marker of infection. Ultimately, these data demonstrate that CM can be infected with very low doses of aerosolized VEEV and develop measurable markers of infection similar to what has been reported for humans [4].

### 3.5. Dose-Dependent Viremia, Fever, and Lymphopenia Pattern Consistent after Aerosol Exposure to VEEV Trinidad Donkey Strain

With the majority of historical data utilizing a very high challenge dose of VEEV TrD, the aim of this phase of the study was twofold: (1) identify the ID_50_ of the VEEV TrD strain and (2) characterize disease after exposure to different doses of the virus (1.0 × 10^1^, 1.0 × 10^3^, 1.0 × 10^5^, and 1.0 × 10^7^ PFU), similar to what was completed with the INH-9813 strain (Table 4). In this study, all 16 NHPs were exposed to VEEV TrD on the same day, ensuring the utilization of exactly the same stock of virus for each animal. Animals were exposed in ascending dose order, with the 1.0 × 10^1^ PFU group exposed first and the 1.0 × 10^7^ PFU group last. Blood samples were collected on 2 d prior to and on days 1–7, 14, 21, and 28 after exposure to evaluate viremia and absolute blood counts. Temperature changes were continuously monitored through implanted telemetry devices. Clinical observations were made prior to exposure to establish baseline behaviors for each animal, and blinded study personnel made observations twice daily after aerosol exposure.

Similar to the viremia pattern observed with increasing doses of INH-9813, the 1.0 × 10^1^ PFU dose group had detectable viremia in only 1 of 4 animals (25%), whereas 4 of 4 animals (100%) had detectable viremia, to varying degrees, in the higher dose groups. Consistent with the pattern observed with the INH-9813 strain, animals exposed to approximately 1.0 × 10^3^ PFU of VEEV TrD reached an overall higher viremia (Figure 5A–D, Figure 6A and Appendix A), with peak titers occurring 48–72 h after exposure (Figure 5B) in comparison to the viremia observed in animals from the higher dose groups (1.0 × 10^5^ and 1 × 10^7^ PFU; Figure 5C and Figure 5D, respectively). Furthermore, using the Wilcoxon exact test to contrast *p*-values for median differences of total viremia in two-tailed analyses identified a statistically significant difference in total viremia between the 1.0 × 10^3^ PFU and 1.0 × 10^7^ PFU groups, *p* = 0.0286 (Appendix A).

Animals exposed to the highest dose of virus consistently had a more robust fever response (Figure 5H), despite having overall lower measured viremia than the 1.0 × 10^5^ and 1.0 × 10^3^ PFU groups (Figure 5G and Figure 5F, respectively). All animals in the two middle dose groups presented with a fever albeit in a less predictable pattern than the 1.0 × 10^7^ PFU group. Use of the Kendell’s Tau (nonparametric) test to evaluate the correlation between total viremia and total fever-hours indicated that these two parameters do not correlate in a statistically consistent manner, R^2^ = 0.3570, *p* = 0.0613 (Appendix A); therefore, total fever-hours should not be used as a surrogate to predict levels of viremia. Pairwise comparison of total fever-hours for each group (Figure 6B) using one-sided Wilcoxon exact tests found each dose group to have significantly lower fever-hours than the next highest dose group (*p* = 0.0143 or *p* = 0.0286), with exception to the 1.0 × 10^3^ to 1.0 × 10^5^ comparison that was likely due to the 1.0 × 10^3^ PFU group having a greater total number of fever-hours than the 1.0 × 10^5^ PFU group (median fever-hours, 84.3 vs. 54.4, respectively).

Evaluation of the complete blood cell counts after VEEV TrD exposure found that each NHP, regardless of exposure dose, had some reduction in total lymphocyte counts; however, clinical lymphopenia (≥30% reduction in total lymphocytes) was not observed in any NHPs in the 1.0 × 10^1^ PFU group (Figure 5I). Similar to observations with INH-9813, lymphopenia was observed within 24 h after exposure to VEEV TrD in 4 of 4 animals in the 1.0 × 10^7^ PFU group and 2 of 4 animals in the 1.0 × 10^5^ PFU group (Figure 5K and Figure 5L, respectively). Interestingly, while 4 of the 4 NHPs in the 1.0 × 10^3^ PFU group had at least 1 d of lymphopenia (Figure 5J), only 2 of the 4 NHPs in the 1 × 10^5^ PFU group had reductions in lymphocytes significant enough to be labeled lymphopenia (Figure 5K). A comparison of the average percent change in lymphocytes for each group revealed a temporal shift in peak lymphocyte reductions dependent on the exposure dose (Figure 6C).

Using viremia as a virus-specific indicator of infection, the ID_50_ of VEEV TrD aerosol exposure to NHPs was identified to be 6.7 PFU by probit analysis or 6.8 PFU by logistic regression. The use of host immune responses as an indication of infection was considered. The ID_50_ of VEEV TrD is 35 PFU by probit analysis of fever-hours or 130 PFU by probit analysis of lymphopenia.

### 3.6. Clinical Observations of VEEV Trinidad Donkey Disease across Exposure Doses

At the time of study execution, efforts were made within the institute to standardize the clinical observation data captured. As a result, only a subset of clinical observation data was collected for this study in comparison to the VEEV INH-9813 studies (Table 5).

Dehydration was noted in 100% of the animals exposed to VEEV TrD. Tremors, predominately of the extremities, were observed in 75% of the animals and, like VEEV INH-9813, were most notable during movement. Lymphadenopathy was noted in 50% of the NHPs during physical examination, a lower proportion of exposed animals than observed after VEEV INH-9813 exposure. Lethargy or a subdued demeanor was noted in just under half (44%) of the NHPs exposed to VEEV TrD, compared to 57% of the NHPs exposed to VEEV INH-9813. It is possible that differences in observed clinical manifestations after VEEV TrD or INH-9813 exposure were a result of the variance in the number of NHPs in each challenge dose.

## 4. Discussion

The development of an effective vaccine or therapeutic against VEEV continues to be of the utmost importance for military personnel and public health protection. A key gap in the progress towards MCM development is a comprehensive understanding of VEEV disease in an animal model that could be used for the licensure of a product under the FDA Animal Rule. Here, we identified the lowest infectious dose of the INH-9813 and TrD strains (subtype IC and IAB, respectively) of VEEV and characterized disease in the CM model at multiple exposure doses (1.0 × 10^1^ through 1.0 × 10^7^ PFU). A strength of these studies was the utilization of statistically powered groups to describe similarities and differences across the exposure doses.

Similar to disease outcomes in humans, VEEV infection of CM results in a nonlethal disease. As a result, identifying reliable, consistent markers of infection in the CM model that can be utilized to determine the efficacy of a MCM is needed. Viremia, measured either by plaque assay or RT-PCR, is the only virus-specific marker of infection currently available. Others have described viremia as transient in the VEEV CM model [36]. However, here, consistent measurement of viremia in animals exposed to doses above 1.0 × 10^3^ PFU was demonstrated. The transient nature of Ma et al.’s findings likely stems from the reduced blood sampling schedule utilized in comparison to the studies reported here, in which we sampled daily for the first seven days after exposure. By sampling daily, we were able to not only identify the day peak viremia is achieved but also describe how it changes based on viral exposure dose. By utilizing statistically powered exposure doses, we were also able to demonstrate the variability observed in viremia curves even when animals were exposed to the same target dose. This becomes important to understand when designing MCM evaluation studies because viremia will likely be a primary end point to demonstrate efficacy. Moreover, viremia has been described as a confounding issue in one vaccine study [37], as inconsistent viremia, that is, the inability to detect viremia in every animal, in the control group made it difficult to determine the efficacy of the vaccine tested. Based on the data presented here, methods to improve detection of viremia include: (1) increasing blood sampling in the first week after exposure, (2) ensuring a statistically powered number of animals in control groups, and (3) choosing a challenge dose that provides the highest titer viremia to ensure measurement is achieved.

Through these studies, an interesting pattern emerged in which exposure to higher viral doses (1.0 × 10^5^–1.0 × 10^7^ PFU) resulted in the rapid development of a lower magnitude of serum viremia; in comparison, the lower 1.0 × 10^3^ PFU dose of virus resulted in a delay in viremia onset but ultimately reached a higher overall titer. This phenomenon may exist for other alphaviruses, as patterns of higher titer viremia have been noted in NHPs aerosol exposed to chikungunya virus (CHIKV) [1] and eastern equine encephalitis virus (EEEV; Appendix A). For CHIKV, NHPs aerosol exposed to 1.0 × 10^4^ PFU had higher viremia values on days 2 and 4 after exposure (on day 6, they had levels below the lower limit of detection [LLOD]) in comparison to those exposed to 1.0 × 10^6^ PFU. In studies we performed with EEEV, viremia levels from NHPs aerosol exposed to doses of the EEEV V105-00210 strain ranging from 6.8 × 10^4^ PFU to 6.9 × 10^7^ PFU were compared [38]. Higher frequencies and magnitudes of viremia were detected in animals exposed to lower doses of virus when compared to levels of viremia in NHPs exposed to higher doses. Indeed, viremia was only detected in two of four animals exposed to the highest doses (~5.0 × 10^7^ PFU). Together, these data suggest exposure to lower doses of alphaviruses (≤1.0 × 10^5^ PFU) results in an increase (magnitude and number of days) in measurable viremia, which suggests researchers should shift from the historical ideal that more viruses are better for model development. We hypothesize that exposure of NHPs to high doses of alphaviruses rapidly stimulates an innate antiviral response that ultimately results in the development of a lower viremia than what is present when animals are exposed to lower, more human exposure-relevant [4] doses of virus. Furthermore, studies by Smith et al. have demonstrated that VEEV infection by the natural, mosquito-vectored route occurs at very low inoculation doses (~11 PFU) [39]. Lastly, infection of human volunteers with VEEV vaccine strain TC-83 found viremia to occur at the earliest two days after inoculation and spanning through days 9–12 after inoculation, more in line with the timing observed in the lower dose inoculated CMs [40]. Therefore, efforts to develop a model system that utilizes lower inoculation doses than previously studied would more likely recapitulate natural VEEV disease.

After a natural infection, VEEV disease has been described as a biphasic illness that initially is a lymphotropic infection that can develop into a neurotropic infection if the virus breaches the CNS [41]. Biphasic fever has been reported in 4 of 17 human cases of aerosol-acquired VEEV disease [26]. As more sophisticated ways of monitoring physiological responses have progressed, a theory has emerged that the febrile response in NHPs mirrors the initial biphasic disease previously described in humans [28,34,42]. This theory states that the initial fever observed in VEEV-infected NHPs corresponds with the onset of the lymphotropic phase of infection (0.5–2 d post-exposure) and is relatively short-lived, lasting only 12–36 h. The second febrile peak then corresponds with viral invasion into the CNS, occurring from 2.5 to 8 d post-exposure [42]. Previous studies report the presence of a biphasic febrile response after VEEV aerosol exposure [21,27,28,34,42], leading us to examine the biphasic febrile response in statistically powered studies of the ability to use fever as an efficacy study endpoint or trigger to treat. We found that while some CMs did display a biphasic febrile response, this fever pattern was not universally observed in all infected animals. In comparison to previous studies, differences in the fever patterns of NHPs in our study may be a result of differences in data analysis methodology and sample size. In the vaccine studies, the data presented is grouped by vaccination status [21,27]. This grouping of datasets may smooth out the variability observed when comparing individual animals, as was done in these studies. To use the febrile response as either an efficacy endpoint or trigger-to-treat marker, methods of temperature data analysis should be universally defined so that data across studies and research institutes can be compared. Additionally, real-time temperature analysis would need to occur to determine when values return to baseline levels for the purpose of a trigger-to-treat determination.

A VEEV expression system is available to produce viral stocks from a DNA plasmid [43,44]. The use of clone-derived stocks provides an advantage over traditional virus passing for stock production as it allows a user to start with the same DNA plasmid each time a stock is made. However, this limits but does not eliminate sequence variance in the stock. Proponents of clone-derived stocks cite the accumulation of potentially attenuating mutations of alphaviruses that result from the serial passage of viruses through cell culture [45,46]. A limitation of these studies is that they only evaluated the effect of cell culture adaptations on lethality in the laboratory mouse, a highly susceptible model where very low exposure doses result in lethality [47]. During the natural enzootic lifecycle, VEEV is maintained in a rodent reservoir with little to no mortality [48]. For this reason, reliance on the laboratory mouse alone as an indicator of VEEV pathogenicity may be misguided. The greatest disadvantage to the use of clone-derived stocks is the lack of sequence diversity that occurs during replication and is observed in nature [15,29,49].

Here, we evaluated several indicators of VEEV disease after various exposure doses to two subtypes (IAB and IC) of biologically derived VEEV. The stocks used in these studies were well-characterized, having a known passage history, low endotoxin levels, no contaminants detected by deep sequencing, and being free of mycoplasma. We evaluated these biologically derived stocks in the CM model and identified the ID_50_ as 12 PFU for the INH-9813 strain and 6.9 PFU for the Trinidad donkey strain based on our ability to measure viremia. The low infectious dose determined in our studies suggests that attenuation of the biologically derived stocks was not observed in the CM model. It is possible that the perceived attenuation of biologically derived virus stocks in the extremely sensitive VEEV mouse model does not translate into a reduction of infectivity in the nonlethal CM model of VEEV disease. Thus far, the lowest reported doses evaluated with clone-derived stocks are 6–6.9 log_10_ PFU [36]. Therefore, more work is needed to determine if clone-derived viral stocks would have a lower ID_50_ in CM.

The objectives of these studies were to define the ID_50_ for two subtypes of VEEV in CMs and identify a challenge dose that reproducibly results in apparent signs of infection (viremia, fever, and lymphopenia) that could be used for MCM development. We found that both IC and IAB VEEV subtypes have very low ID_50_ doses (<15 PFU) based on the ability to detect viremia after challenge. By increasing the number of animals at a given target dose, we were able to establish a challenge dose (1.0 × 10^3^ PFU) that resulted in measurable viremia in every animal exposed, thereby supplying a virus-specific marker of infection that could be utilized as an efficacy endpoint. While these studies were statistically powered to support this finding, additional studies are needed to examine this lower challenge dose in the context of MCM efficacy studies.

## Figures and Tables

**Figure 1 viruses-15-02351-f001:**
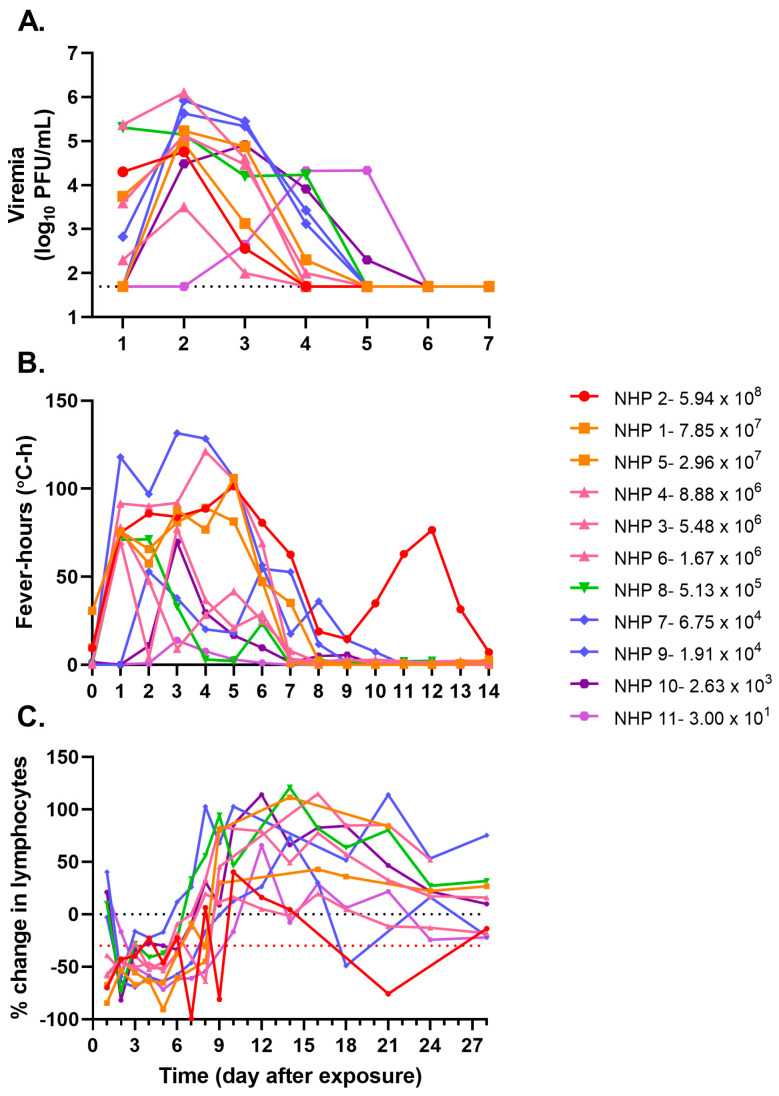
VEEV INH-9813 dose ranging study. Cynomolgus macaques were head-only aerosol exposed to VEEV. (**A**) Individual animal viremia. The blood was collected daily for the first 7 days after aerosol exposure. Viremia in diluted whole blood was measured using a plaque assay on Vero 76 cells. The dotted line indicates the lower limit of detection for the assay. (**B**) Daily fever-hours measured by continuous telemetry implants. Baseline data were collected prior to exposure to define each animal’s normal circadian patterns. After exposure, temperatures were compared to the baseline values to calculate fever-hours. For NHP 12, a peak in fever-hours was observed later in infection, resulting from a delayed biphasic fever response in that animal (Appendix A). (**C**) Percent change in absolute lymphocyte counts after VEEV exposure. At least two samples were collected prior to challenge to establish baseline values. Lymphopenia, defined as a ≥30% decrease in absolute lymphocyte values compared to baseline, is indicated by the red dotted line.

**Figure 2 viruses-15-02351-f002:**
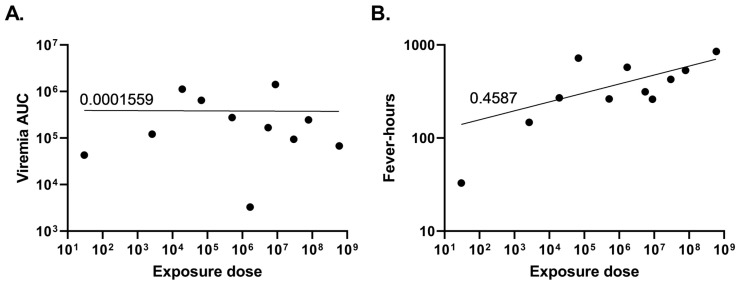
Viremia and fever in relation to VEEV exposure dose. (**A**) Total viremia (as measured by the area under the curve) or (**B**) total fever-hours over the course of 28 days were graphed as a function of the dose to which the animals were exposed. Nonlinear fit tests were conducted. The goodness-of-fit value for viremia is R squared 0.0001559, and for fever-hours, it is R squared 0.4587.

**Figure 3 viruses-15-02351-f003:**
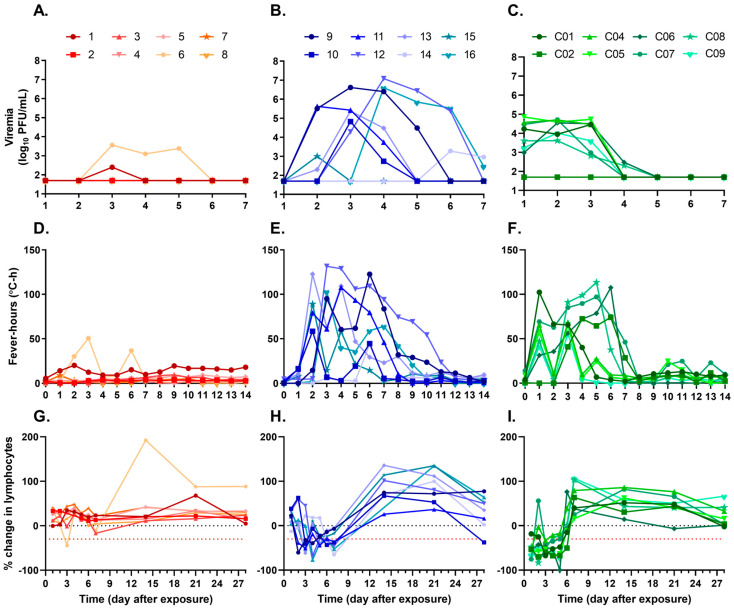
Refinement of VEEV INH-9813 disease markers after low, medium, or high exposure doses. Cynomolgus macaques were head-only aerosol exposed to VEEV. Viremia by animal after exposure to a (**A**) low 1.0 × 10^1^ PFU, (**B**) medium 1.0 × 10^3^ PFU, or (**C**) high 1.0 × 10^6^ PFU dose. Blood was collected daily for the first 7 days after aerosol exposure. Viremia in diluted whole blood was measured by a plaque assay on Vero 76 cells. Values below the lower limit of detection (LOD) for the assay were set to the LOD for graphing. Daily fever-hours by animal measured by continuous telemetry implants after exposure to a (**D**) low, (**E**) medium, or (**F**) high dose. Baseline data were collected prior to exposure to define each animal’s normal circadian patterns. After exposure, temperatures were compared to the baseline values to calculate fever-hours. Percent change in absolute lymphocyte counts after exposure to a (**G**) low, (**H**) medium, or (**I**) high dose. At least two samples were collected prior to the challenge to establish baseline values. Lymphopenia, defined as a ≥30% decrease in absolute lymphocyte values compared to baseline, is indicated by the red dotted line.

**Figure 4 viruses-15-02351-f004:**
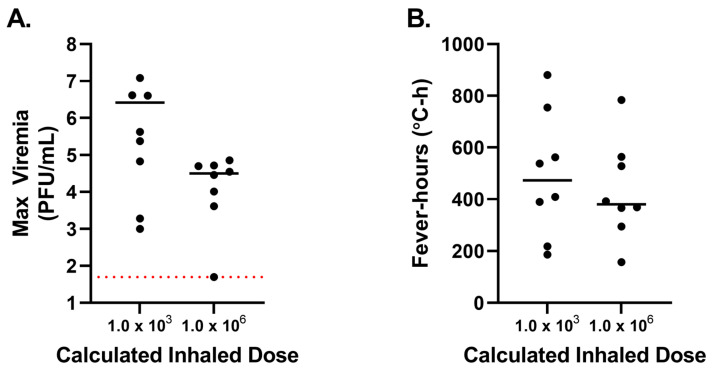
Peak viremia and total fever-hours for cynomolgus macaques exposed to medium or high doses of VEEV INH-9813. Animals were exposed to a target inhaled dose of 1.0 × 10^3^ PFU or 1.0 × 10^6^ PFU of VEEV. (**A**) Peak viremia titers measured by plaque assay on Vero 76 cells and (**B**) total fever-hours per animal are compared. The lines represents the mean values for each dose group. The red dotted line indicates the lower limit of detection (LOD) for the assay. Values below the LOD for the assay were set to the LOD for graphing.

**Figure 5 viruses-15-02351-f005:**
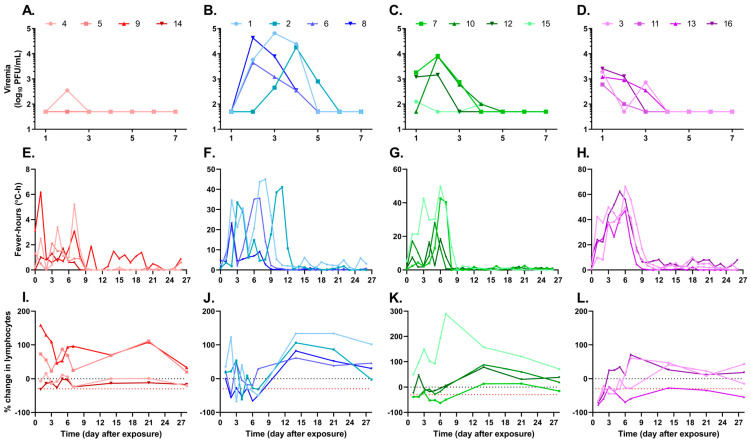
Dose-dependent viremia, fever, and lymphopenia pattern consistent after aerosol exposure to VEEV Trinidad donkey strain. Cynomolgus macaques were head-only aerosol exposed to VEEV. Viremia by animals after exposure to a (**A**) 1.0 × 10^1^ PFU, (**B**) 1.0 × 10^3^ PFU, (**C**) 1.0 × 10^5^ PFU or (**D**) 1.0 × 10^7^ PFU dose. Blood was collected daily for the first 7 days after aerosol exposure. Viremia in serum was measured by a plaque assay on Vero 76 cells. Values below the lower limit of detection (LOD) for the assay were set to the LOD for graphing. Daily fever-hours measured by continuous telemetry implants after exposure to a dose of (**E**) 1.0 × 10^1^ PFU, (**F**) 1.0 × 10^3^ PFU, (**G**) 1.0 × 10^5^ PFU or (**H**) 1.0 × 10^7^ PFU. Baseline data were collected prior to exposure to define each animal’s normal circadian patterns. After exposure, temperatures were compared to the baseline values to calculate fever-hours. Percent change in absolute lymphocyte counts after exposure to a dose of (**I**) 1.0 × 10^1^ PFU, (**J**) 1.0 × 10^3^ PFU, (**K**) 1.0 × 10^5^ PFU, or (**L**) 1.0 × 10^7^ PFU. At least two samples were collected prior to the challenge to establish baseline values. Lymphopenia, defined as a ≥30% decrease in absolute lymphocyte values compared to baseline, is indicated by the red dotted line.

**Figure 6 viruses-15-02351-f006:**
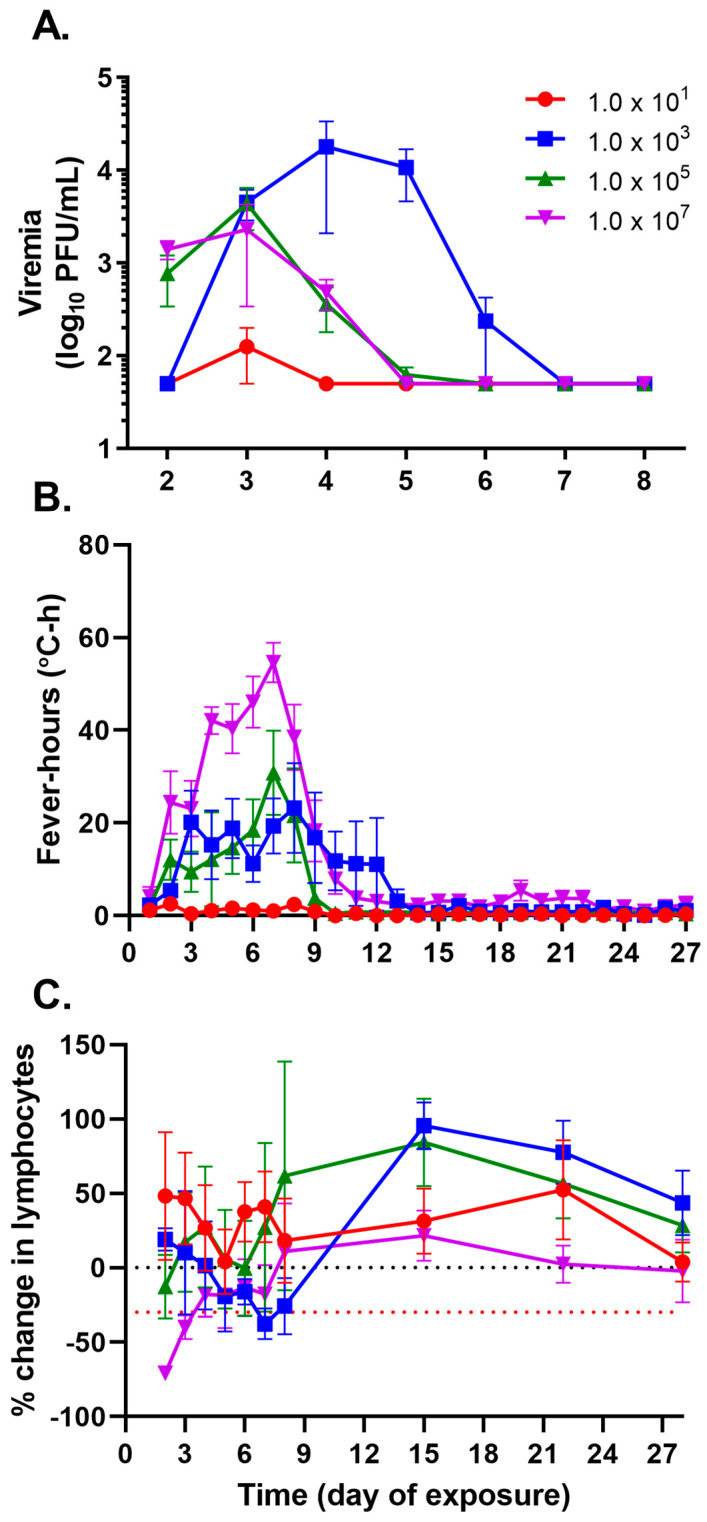
Comparison of viremia, fever, and lymphocyte changes by exposure dose of VEEV TrD. (**A**) Mean (±standard error of the mean) viremia after aerosol exposure to VEEV TrD. Blood was collected daily from each cynomolgus macaque for the first 7 days after aerosol exposure. Viremia in serum was measured by a plaque assay on Vero 76 cells. Values below the lower limit of detection (LOD) for the assay were set to the LOD for graphing. (**B**) Mean (±standard error of the mean) daily fever-hours measured by continuous telemetry implants after aerosol exposure to VEEV TrD. Baseline data were collected prior to exposure to define each animal’s normal circadian patterns. After exposure, temperatures were compared to the baseline values to calculate fever-hours. (**C**) Mean (±standard error of the mean) percent change in absolute lymphocyte counts after aerosol exposure to VEEV TrD. At least two samples were collected prior to the challenge to establish baseline values. Lymphopenia, defined as a ≥30% decrease in absolute lymphocyte values compared to baseline, is indicated by the red dotted line.

**Table 1 viruses-15-02351-t001:** Summary of VEEV INH-9813 staircase study.

NHP ID	Sex	Calculated Inhaled Dose (PFU)	Viremia (AUC)	Max Viremia (PFU/mL)	Number of Days Viremic (PA)	Fever-Hours (>3 SD)	Number of Days of Lymphopenia (Range 1–7)	PRNT80
2	M	5.94 × 10^8^	67,542	5.7 × 10^4^	3	849.3	5	ND
1	F	7.85 × 10^7^	244,350	1.7 × 10^5^	3	531.5	6	ND
5	M	2.96 × 10^7^	94,015	9.0 × 10^4^	3	426.4	6	2560
4	M	8.88 × 10^6^	1,407,875	1.3 × 10^6^	3	260.2	6	2560
3	M	5.48 × 10^6^	166,400	1.4 × 10^5^	4	313.1	5	2560
6	M	1.67 × 10^6^	3275	3.2 × 10^3^	3	575.3	4	2560
8	M	5.13 × 10^5^	274,325	2.0 × 10^5^	4	263	3	2560
7	M	6.75 × 10^4^	642,840	4.2 × 10^5^	4	719.0	1	1280
9	M	1.91 × 10^4^	1,114,170	8.3 × 10^5^	3	269.2	6	1280
10	M	2.63 × 10^3^	120,520	8.2 × 10^4^	4	146.9	3	2560
11	M	3.00 × 10^1^	42,800	2.2 × 10^4^	3	32.8	5	2560

PFU = plaque-forming units, AUC = area under the curve, PA = plaque assay, SD = standard deviation, PRNT = plaque reduction neutralization test, ND = not done; coloring in the Viremia, Max Viremia, and Fever-hours columns shows a gradient from lowest values (green) to highest values (red).

**Table 2 viruses-15-02351-t002:** Summary of the VEEV INH-9813 refinement study.

NHP ID	Sex	Calculated Inhaled Dose (PFU)	Viremia (AUC)	Max Viremia (PFU/mL)	Number of Days Viremic (PA)	Number of Days Viremic (RT-PCR)	Fever-Hours (>3 SD)	Number of Days of Lymphopenia (Range 1–7)	PRNT80
2	F	1.8 × 10^1^	0	5.0 × 10^1^	0	4	75.7	0	<20
1	F	2.0 × 10^1^	200	2.5 × 10^2^	1	3	341.4	0	<20
3	F	2.1 × 10^1^	0	5.0 × 10^1^	0	2	109.20	0	<20
7	M	2.7 × 10^1^	0	5.0 × 10^1^	0	0	65.6	0	<20
4	F	2.7 × 10^1^	0	5.0 × 10^1^	0	2	70.3	0	<20
8	M	3.2 × 10^1^	0	5.0 × 10^1^	0	1	60.2	0	<20
6	M	3.6 × 10^1^	7100	3.6 × 10^3^	3	5	169.6	1	>5120
5	M	6.1 × 10^1^	0	5.0 × 10^1^	0	3	181.9	0	<20
**Group** **Average**		**3.03 × 10^1^**	**912.5**	**5.2 × 10^2^**	**0.5**	**2.5**	**134.2**	**0**	**-**
9	F	1.1 × 10^3^	6,950,300	4.1 × 10^6^	4	6	562.4	3	>5120
15	M	1.3 × 10^3^	950	1.0 × 10^3^	1	6	389.9	1	>5120
12	M	1.5 × 10^3^	14,953,800	1.2 × 10^7^	4	5	880.1	3	>5120
13	M	1.6 × 10^3^	265,050	2.4 × 10^5^	3	6	754.4	2	>5120
10	F	1.7 × 10^3^	67,450	6.7 × 10^4^	2	5	217.7	3	>5120
14	M	1.8 × 10^3^	2275	1.9 × 10^3^	2	4	186.3	2	>5120
16	M	1.8 × 10^3^	5,044,975	4.0 × 10^6^	4	4	408.4	3	>5120
11	F	1.9 × 10^3^	690,300	4.2 × 10^5^	3	6	537.5	4	>5120
**Group** **Average**		**1.6 × 10^3^**	**3,496,888**	**2.6 × 10^6^**	**3**	**5**	**492.1**	**3**	**-**
C07	M	9.1 × 10^5^	67,075	5.2 × 10^4^	3	6	783.6	4	10,240
C06	M	9.3 × 10^5^	67,600	3.5 × 10^4^	4	5	391.7	3	20,480
C04	M	9.4 × 10^5^	96,875	5.0 × 10^4^	3	7	368.9	5	10,240
C01	M	1.1 × 10^6^	45,605	2.9 × 10^4^	3	7	528.1	6	20,480
C05	F	1.1 × 10^6^	127,875	7.1 × 10^4^	3	6	294.2	5	10,240
C02	F	1.3 × 10^6^	0	5.0 × 10^1^	0	3	365.9	3	20,480
C09	F	1.8 × 10^6^	14,900	1.0 × 10^4^	3	6	156.8	3	10,240
C08	F	2.5 × 10^6^	6755	4.1 × 10^3^	4	7	563.6	5	10,240
**Group** **Average**		**1.3 × 10^6^**	**53,336**	**3.1 × 10^4^**	**3**	**6**	**431.6**	**4**	

PFU = plaque-forming units, AUC = area under the curve, PA = plaque assay, SD = standard deviation, PRNT = plaque reduction neutralization test; coloring in the Viremia, Max Viremia, and Fever-hours columns shows a gradient from lowest values (green) to highest values (red).

**Table 3 viruses-15-02351-t003:** Observed signs and frequency of VEEV disease across all NHPs exposed to VEEV INH-9813 (n = 35).

Observed Signs	Number	Percentage
Tremors	33	94%
Lymphadenopathy	32	91%
Loss of appetite	22	63%
Subdue/lethargy	20	57%
Alterations to fur (ungroomed, piloerection, etc.)	13	37%
Dehydration	12	34%
Nystagmus	11	31%
Hyperactive/reactive	9	26%
Yawning/jaw movements	7	20%
Teeth bearing	6	17%
Depressed/sad face	6	17%
Aggressive/agitated	6	17%
Licking cage	4	11%
Hiding back of cage	3	9%
Vomiting	3	9%
Diarrhea	2	6%
Photophobia	2	6%
Legs curled up underneath NHP	2	6%

NHP = nonhuman primate.

**Table 4 viruses-15-02351-t004:** Summary of the VEEV Trinidad donkey strain bridging study.

NHP ID	Sex	Calculated Inhaled Dose	Viremia (AUC)	Max Viremia (PFU/mL)	Number of Days Viremic (PA)	Number of Days Viremic (RT-PCR)	Fever-Hours (>3 SD)	Number of Days of Lymphopenia (Range 1–7)	PRNT80
4	M	7.30 × 10^0^	300	3.50 × 10^2^	1	0	16	0	<10
5	M	6.20 × 10^0^	0	5.00 × 10^1^	0	0	10.4	0	<10
9	F	5.10 × 10^0^	0	5.00 × 10^1^	0	0	29.1	0	<10
14	F	3.40 × 10^0^	0	5.00 × 10^1^	0	0	6.7	1	<10
**Group** **Average**		**5.50 × 10^0^**	**75.00**	**1.25 × 10^2^**	**0.25**	**0.00**	**15.55**	**0.25**	**-**
1	M	1.10 × 10^3^	95,650	6.60 × 10^4^	3	6	289.4	3	20,480
2	M	1.00 × 10^3^	19,025	1.80 × 10^4^	3	4	216.3	2	10,240
6	F	4.90 × 10^2^	5800	4.40 × 10^3^	3	3	157.5	1	20,480
8	F	1.20 × 10^3^	50,550	7.90 × 10^3^	3	4	70.2	4	20,480
**Group** **Average**		**9.48 × 10^2^**	**42,756.25**	**2.41 × 10^4^**	**3.00**	**4.25**	**183.35**	**2.50**	**-**
7	F	4.60 × 10^5^	9700	8.20 × 10^3^	3	5	260.9	0	20,480
10	F	2.30 × 10^5^	8350	7.80 × 10^3^	3	4	124.5	6	10,240
12	M	1.40 × 10^5^	1975	1.50 × 10^3^	2	3	92.7	2	10,240
15	M	2.30 × 10^5^	87.5	1.30 × 10^2^	3	4	70.2	0	10,240
**Group** **Average**		**2.65 × 10^5^**	**5028.13**	**4.41 × 10^3^**	**2.75**	**4.00**	**137.08**	**2.00**	**-**
3	M	1.60 × 10^7^	1613	1.90 × 10^3^	2	4	436	3	10,240
11	M	3.80 × 10^7^	325	6.00 × 10^2^	2	2	303	4	20,480
13	F	2.50 × 10^7^	1725	1.20 × 10^3^	3	6	232.3	5	20,480
16	F	2.60 × 10^7^	2450	8.20 × 10^3^	2	2	417.4	2	10,240
**Group** **Average**		**2.63 × 10^7^**	**1528.25**	**2.98 × 10^3^**	**2.25**	**3.50**	**347.18**	**3.50**	**-**

PFU = plaque-forming units, AUC = area under the curve, PA = plaque assay, SD = standard deviation, PRNT = plaque reduction neutralization test; coloring in the Viremia, Max Viremia, and Fever-hours columns shows a gradient from lowest values (green) to highest values (red).

**Table 5 viruses-15-02351-t005:** Observed signs and frequency of VEEV disease across all NHPs exposed to VEEV TrD (n = 16).

Observed Signs	Number	Percentage
Tremors	12	75%
Lymphadenopathy	8	50%
Subdue/lethargy	7	44%
Dehydration	16	100%

## Data Availability

The data presented in this study are provided within the article, presented as Appendix A, and available on request from the corresponding author.

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
