# Peer review of "Defining the Cynomolgus Macaque (Macaca fascicularis) Animal Model for Aerosolized Venezuelan Equine Encephalitis: Importance of Challenge Dose and Viral Subtype"

_viruses, 2023, doi:10.3390/v15122351_

Round 1
Reviewer 1 Report
Comments and Suggestions for Authors
The manuscript well written and comprehensive. The multiple studies provide data and the analyses to establish a well-defined nonhuman primate model for VEEV. This is critical for testing medical countermeasures under the FDA Animal Rule.
Major comments:
Line 148: The telemetry implants monitor both temperature and activity data. However, the activity data is not included in the manuscript.
Understandably the focus is on changes in total lymphocyte counts in the VEEV NHP model. However, since complete blood counts were determined for each time point, information on other hematological parameters is available and could be added to the manuscript in order to more fully characterize the model.
Line 198 in the methods indicates that plaque assays were performed on nasal swabs, throat swabs, and tissues in addition to blood. Also, line 235 indicates that viral RNA was extracted from throat swabs, nasal swabs, and tissues. However, the data is not included in the manuscript.
Line 604 – Figure S7 presented data for NHPs aerosol exposed to EEEV. However, since the manuscript does not provide information on the EEEV study, the figure should be removed from this manuscript and a reference added.
Minor comments:
The figure legend for Figure S1 should indicate the challenge virus.
Line 372--373 indicates that 3 NHPs had detectable viremia for up to 4 days but includes only a single value for each NHP and does not indicate which day for the values.
Line 563 – suggest deleting “a”.
Line 570 – suggest replacing important with importance.
Figure S6 – suggest removing the protocol, project, and analyst. SAS and version# can be added to the legend.
Figure S6A – indicate in the legend that the line in the box and whisker plot represents the median +/- confidence interval.
Figure S6C – label the y-axis.
Author Response
The Authors appreciate the Reviewer's time and attention to provide a thorough and helpful review. The Authors have addressed the items suggested the by Reviewer where the suggestions to improve the manuscript messaging. Below are the responses to specific comments.
Line 148: The telemetry implants monitor both temperature and activity data. However, the activity data is not included in the manuscript. While the implants record temperature and activity data, only temperature data was analyzed. As a result, we elected to remove the “and activity” portion of the statement.
Understandably the focus is on changes in total lymphocyte counts in the VEEV NHP model. However, since complete blood counts were determined for each time point, information on other hematological parameters is available and could be added to the manuscript in order to more fully characterize the model. The Authors acknowledge that including the CBC data for each timepoint would more fully characterize the model; however, lymphopenia is the most common hematological parameter observed in human VEEV infection and therefore the Authors want to highlight its presence in the NHP model. All other hematological parameters were assessed by the statistician, but none rose to the level of statistical significance. To maintain as streamlined a story as possible, the Authors are electing to keep the CBC results focused on the lymphocyte counts and lymphopenia.
Line 198 in the methods indicates that plaque assays were performed on nasal swabs, throat swabs, and tissues in addition to blood. Also, line 235 indicates that viral RNA was extracted from throat swabs, nasal swabs, and tissues. However, the data is not included in the manuscript. The Authors thank the reviewer for pointing out this lack of information provided. Swabs were analyzed by plaque assay and that information has been added on line 265-268. We removed the swabs from the RT-PCR methods section as samples were collected but never analyzed.
Line 604 – Figure S7 presented data for NHPs aerosol exposed to EEEV. However, since the manuscript does not provide information on the EEEV study, the figure should be removed from this manuscript and a reference added. The Authors acknowledge the reviewers request to cite the source of Figure S7 or remove it from the manuscript. The Authors believe inclusion of Figure S7 is necessary to provide evidence of a potential alphavirus-wide phenomenon of lower challenge doses resulting in higher viremia. Unfortunately, this data is not yet published so a reference cannot be provided (as indicated in line 611). More details of the study were added to the legend for clarity on study design.
Minor comments:
The figure legend for Figure S1 should indicate the challenge virus. Added virus to figure legend.
Line 372--373 indicates that 3 NHPs had detectable viremia for up to 4 days but includes only a single value for each NHP and does not indicate which day for the values. The Authors understand the reviewer’s comment. The titer listed for each NHP is the calculated inhaled dose, not the viremia titer for each NHP. The Authors see how this could be confusing, so we have removed the calculated inhaled dose value for those NHPs.
Line 563 – suggest deleting “a”. Removed
Line 570 – suggest replacing important with importance. Thank you for this suggestion, change made.
Figure S6 – suggest removing the protocol, project, and analyst. SAS and version# can be added to the legend. The Authors appreciate the suggestion and have made these changes
Figure S6A – indicate in the legend that the line in the box and whisker plot represents the median +/- confidence interval. Updated figure legend to indicate what box and whisker plot represents.
Figure S6C – label the y-axis. Edited y-axis to indicate what it was a measure of.
Thank you for your review. The Authors recognizes the extra time your took to comment on our work and appreciate your efforts.
Reviewer 2 Report
Comments and Suggestions for Authors
The manuscript by Burke and colleagues presents an exceptionally detailed description of experimental VEEV infection in cynomolgous primates. The work was stimulated by lack of a “realistic” challenge model for this species would be useful in validating vaccines or therapeutic countermeasures for this virus. The manuscript is very well written and logically presented and I do not have substantive criticisms or suggestions for improvement. The manuscript is very long and there is considerable redundancy between text and figures, but I simply could not come up with a reasonable way to alleviate that issue. One small item the authors might consider is significant digits. I realize that most of the virus titers mentioned in the text are calculated (i.e., means), but stating that an ID50 is 6.7 PFU (line 23 as an example) seems rather silly and I might suggest considering reporting those values as after rounding to integers (this is not a strong suggestion)
Overall, this manuscript is somewhat of a tour de force for the topic at hand and I found the disconnect between magnitude of viremia and virus dose intriguing.
Author Response
The Authors greatly appreciate the Reviewer's time to evaluate the manuscript. We too recognize that it is a lengthy body of work that can seem somewhat repetitive but could not find another way to streamline the message that lower challenge doses result in higher viremia across multiple VEEV Strains without presenting all the data available to support that claim.
To specifically address the Reviewer comment:
I realize that most of the virus titers mentioned in the text are calculated (i.e., means), but stating that an ID50 is 6.7 PFU (line 23 as an example) seems rather silly and I might suggest considering reporting those values as after rounding to integers (this is not a strong suggestion). The Authors appreciate the Reviewer’s comments. While we agree rounding to integers would make more sense, since these numbers were statistically derived, we prefer to maintain the significant digits identified by Mr. White our statistician.
Reviewer 3 Report
Comments and Suggestions for Authors
The study addresses the lack of knowledge in low dose infection of VEEV in animal models. It showcases the importance in defining a right virus challenge dose as well as the relationship between the challenge dose vs. clinical outcomes using NHP model.
As the data presentations were based on each individual animals, it is hard to follow each line in the graphs. I would suggest using a heat map style presentation to better present data.
Author Response
The Authors would like to thank the Reviewer for taking the time to supply feedback on the manuscript. The Reviewer commented "As the data presentations were based on each individual animals, it is hard to follow each line in the graphs. I would suggest using a heat map style presentation to better present data."
The Authors appreciate the suggestion, but it is unclear as to what dataset exactly the reviewer is referring to that would benefit from presentation as a heat map. For that reason, no changes were made to the style of presentation.